# Electrical Power Diversification: An Approach Based on the Method of Maximum Entropy in the Mean

**DOI:** 10.3390/e23030281

**Published:** 2021-02-26

**Authors:** Rafael Bautista, Henryk Gzyl, Enrique ter Horst, Germán Molina

**Affiliations:** 1School of Management, Universidad de los Andes, Bogotá 111711, Colombia; rbautist@uniandes.edu.co; 2Centro de Finanzas, IESA, Caracas 1011, Venezuela; henryk.gzyl@iesa.edu.ve; 3Idalion Capital Group, Quantitative Trading, 12 Hay Hill, London W1J 8NR, UK; german@germanmolina.com

**Keywords:** electricity supply and demand, energy supply diversification, contamination constraints, inverse problems, maximum entropy in the mean

## Abstract

Electrical energy is generated in different ways, each located at some specific geographical area, and with different impact on the environment. Different sectors require heterogeneous rates of energy delivery, due to economic requirements. An important problem to solve is to determine how much energy must be sent from each supplier to satisfy each demand. Besides, the energy distribution process may have to satisfy ecological, technological, or economic cost constraints.

## 1. Introduction

The problem of optimal and well-diversified energy distribution in a given country is central to the well-functioning of economic and social infrastructures. There are two aspects to the problem. On the one hand, we have to consider the problem of satisfying the demand without unnecessary surplus (unless there is a market for the surplus). On the other hand, we must ensure that the resulting distribution is well-diversified.

The first problem is a typical exercise of demand-supply with given cost constraints. Costs include economic, ecological, and social costs. Any cost that can be translated into a constraint on the energy flow between consumer and supplier (or source and sink) should be taken into account. This problem has received considerable attention in the applied mathematics literature when the data about the amounts produced are exact. A review of the literature is available in [1].

In addition, as there might be uncertainties in the demand and/or the supply, it is realistic to suppose that the data are specified up to intervals. Of course, there are obvious interval constraints of this type: a given supplier cannot produce more than a certain amount at a given site. However, the amount produced may fluctuate within a range within the maximum capacity. Similar limitations can be observed on the demand side.

Linear programming techniques are one of the standard tools to deal with the problem, especially when there is a monetary cost that must be minimized. The main goal of this paper is to propose the method of maximum entropy on the mean to deal with the transportation problem with cost constraints and range constraints on the solutions. The method that we propose is based on an extension of the standard method of maximum entropy proposed in [2]. Attempts to solve the problem using maximum entropy based methods are long-dated (see Chapter 13 in Kapur’s [3] and the references in that chapter for earlier work, in particular to the work by Wilson [4]). An interesting application can be found in [5]. Our approach to the use of the method of maximum entropy is completely different.

The problem of diversification is reviewed in [6,7], where more references are listed, and we bring in the entropy-based measure proposed there to measure the diversification of the solution to the problem considered above.

To state the problems treated here, consider *M* sources of electricity and denote by $\left\{S_{1},\ldots ,S_{M}\right\}$ the amount supplied by each source. The labeling addresses the possibility that in the same geographical area there may exist two different types of electricity sources, which would be labeled by two different subscripts.

Consider *N* different electricity consumers, each of whom requiring $\left\{D_{1},\ldots ,D_{N}\right\}$ units of energy per unit time. These may describe geographical areas and/or different types of consumer within a given area. For example, within a geographical area, there may exist large chemical industries or mineral processing plants that may need large amounts of electricity, besides what is needed for domestic use by the local population.

The first group of constraints is imposed by the nature of the generation process or by the local demand of electricity: We suppose that the demand $D_{i}$ at each site i=1,…,N and the supply $S_{j}$ at site j=1,…,M are both known. Demand at site *i* represents the demand by a type of energy consumer which might be distributed geographically, but which shares some type of collective characteristic, such as being a domestic (household) consumer in a geographic area.

### 1.1. Problem Statements

If there were no other constraints, the simplest problem of distributing the electricity could be stated as follows.

**Problem** **1**(First electricity supply-demand problem). *Denote by $\left\{x_{i,j}:i=1,\ldots ,N;j=1,\ldots ,M\right\}$ the amount of electricity required at site i and produced at site j per unit time. We suppose that the network is such that the every electricity-producing site is connected to every consuming site. Otherwise, we set the corresponding $x_{i,j}=0$ without further ado.*
*The electricity matching problem consists of determining values $\left\{x_{i,j}:i=1,\ldots ,N;j=1,\ldots ,M\right\}$ that satisfy the following demand-supply constraints:*
(1)$$\sum_{j=1}^{M}x_{i,j}=D_{i},\;\;\;for\;\;i=1,\ldots ,N\;\;\;and\;\;\;\sum_{i=1}^{N}x_{i,j}=S_{j}\;\;for\;\;j=1,\ldots ,M.$$



**Comments:**
(**i**)Production per unit time refers to an average produced or required during some standardized time interval (one hour, for example).(**ii**)These constraints can be replaced by intervals to allow for uncertainty in the demand or uncertainty in the supply. We describe this further below.(**iii**)Note that we are taking into account possible nonlinear constraints, resulting from the actual physical transport of energy through the network, in which the losses could depend on the amount of energy being transported.


In practice, cost constraints may also exist. Here, we suppose that costs are regulated and fixed (by competition or government agencies) and fairly passed on to consumers, but these constraints have to be taken into account.

There might also exist regulatory environmental constraints. Each mode of electricity generation has an environmental impact, measured by $C_{i}$ grams of contaminant per unit of electricity generated by the *i*th supplier per unit time. Therefore, the total amount of contaminants generated by the supplier is
$$\sum_{i=1,j=1}^{i=N,j=M}C_{i}x_{i,j}=\sum_{i=1}^{N}C_{i}\left(\sum_{j=1}^{M}x_{i,j}\right).$$

With this, instead of Problem 1, we consider now the following:

**Problem** **2**(Second electricity supply-demand problem). *With the notation introduced above, the electricity matching problem consists of determining $\left\{x_{i,j}:i=1,\ldots ,N;j=1,\ldots ,M\right\}$ that satisfy the demand-supply constraints (1) as well as cost constraints solving:*
(2)$$\sum_{i=1,j=1}^{i=N,j=M}C_{r,(i,j)}x_{i,j}=K_{r},\;\;\;r=1,\ldots ,R.$$

Here, $K_{r}$ is a maximum cost (environmental impact) generated (or incurred) by the electricity-producing system that regulators allow. Most of the time the constraint will not depend on the connection (edge) (i,j) but only on the source (type of energy that is produced at the *j*th source). In this case, $C_{r,(i,j)}$ does not change as *i* changes.

For a combination of technological and economical reasons, we might be forced to consider flexible constraints. For example, to cover for possible downward or upward movement in demand or supply, we might consider those values to fall in a range. Similarly, we could include a tolerance for sudden fluctuations in the cost of production, or the demand at a certain node. Instead of point-valued constraints, we might replace both Problems 1 and 2 by:

**Problem** **3**(Third electricity supply-demand problem). *To allow for different types of constraints, we extend our previous notation. Denote by C a R×d-matrix, where d=N×M, and denote its elements by $C_{r,(i,j)}$, where 1≤r≤R labels the cost constraints (or cost restrictions). Now, instead of a point value K, we consider a range dataset given by:*
(3)$$K=\prod_{r=1}^{R}\left[K_{1}^{r},K_{2}^{r}\right],\;\;\;with\;\;\;K_{1}^{r}\leq K_{2}^{r}\;\;for\;\;\;r=1,\ldots ,R.$$*The problem to solve is to find $x_{i,j}$ in some given range such that:*
(4)Cx∈K.
*When K reduces to a point, that is, when K={k} with $k\in R^{R},$ we have a problem with*
**point constraints**.


Problems 1–3 are ill-posed linear inverse problems subject to convex constraints. As such, each might have infinitely many possible solutions because the number of unknowns is usually significantly larger than that of the dataset. The convex constraints include positivity constraints as well as those in (3).

The method of maximum entropy in the mean is especially designed to deal with this type of problems. The representation of the solution is such that it allows for explicit sensitivity analysis. The standard maximum entropy method (SME) is the stepping stone towards the method of maximum entropy in the mean (MEM).

### 1.2. Contents of the Paper

Building on the aforecited notation, we briefly describe the solution to Problems 1–3 in Section 2. The solution of the three problems, given by (21), looks the same, although what changes is the specification of the problem data. The actual derivation of the solutions of Problems 1 and 2 follows the same pattern, and Problem 3 uses the solution of Problem 2 as a stepping stone.

In Section 3, we recall the mathematical details of the procedure to arrive at the results listed in Section 2. We include a short digression on using the concept of entropy to quantify how diversified is a solution to the supply-demand problem. Further, we explicitly compute it for each of the examples.

In Section 4, we work out two illustrating toy examples which take into account all the essentials. In the first one, we consider only point data, while, in the second example, we show the flexibility of the approach with some of the data in intervals. The natural production constraints are specified up to an interval to incorporate the possibility of fluctuation in the energy generation output.

## 2. Mathematical Model Derivation

Here, we collect the basics results about the SME method, MEM with point data, and MEM with data in ranges. We explain the resulting (21).

### 2.1. MEM for Point Data

The standard maximum entropy (SME) method was almost simultaneously proposed by Jaynes [2] and Kullback [8]. It is a variational method to determine a probability density ρ(ξ) such that dP(ξ)=ρ(ξ)dQ(ξ) satisfying some integral constraints. MEM focuses on how to transform an ill-posed linear inverse problem with constraints upon the solutions into a problem of determining a probability density. This is explained in [9,10]. To continue, we need to introduce some notations.

Let $\Omega =\prod_{i=1}^{d}\left[a_{i},b_{i}\right]$ and denote by F the Borel subsets of Ω. Consider a (reference) measure dQ(ξ) upon F. Let P(Q)={Pprobability measure on Ω:dP(ξ)=ρ(ξ)dQ(ξ)}, that is the class of probabilities absolutely continuous with respect to Q. Let X:Ω→Ω denote the coordinate or identity mapping, that is X(ξ)=ξ. Let C be a K×d-matrix and let $y\in R^{K}$ be some given vector. The constraint set is defined by:(5)$$C\left(Q,y\right)=\{P\in P|\;CE_{P}\left[X\right]=y\}.$$This is a convex set. The best way to choose a point in it is to maximize some meaningful concave function defined on it.

**Definition** **1.**
*The entropy of P∈P is defined by*
(6)$$S_{Q}\left(P\right)=-\int_{\Omega }\rho \left(\xi \right)\ln\rho \left(\xi \right)dQ\left(\xi \right).$$
*whenever $\int_{\Omega }\rho \left(\xi \right)\left|\ln\rho \left(\xi \right)\right|dQ\left(\xi \right)<\infty ,$ and equal to −∞ otherwise.*


Then, we can state the entropy maximization problem:

**Problem** **4.**
*Find $\rho^{*}\left(\xi \right)$ at which $S_{Q}\left(P\right)$ achieves its maximum subject to the constraints (5). T hat is, determine*
$$\rho^{*}=argsup\left\{S_{Q}\left(P\right)\;|\;P\in C\left(Q,y\right)\right\}.$$


Once $dP*=\rho^{*}dQ$ has been determined, the solution to the algebraic problem is given by
(7)$$x^{*}=E_{P^{*}}\left[X\right]=\int_{\Omega }\xi \rho^{*}\left(\xi \right)dQ\left(\xi \right)$$
which clearly satisfies $x^{*}\in \Omega$ and $Cx^{*}=y.$ In our present case, a direct application of the method of Lagrange multipliers leads to the representation
(8)$$\rho^{*}\left(\xi \right)=\frac{e^{-\langle C^{t}\lambda^{*},\xi \rangle }}{Z(\lambda^{*})}$$
where the normalization factor is given by
(9)$$Z\left(\lambda \right)=\int_{\Omega }e^{\langle C^{t}\lambda ,\xi \rangle }dQ\left(\xi \right),\;\;\;\lambda \in R^{K}.$$The optimal Lagrange multiplier $\lambda^{*}$ is to be found minimizing the dual entropy function
(10)Σ(λ,y)=lnZ(λ)+〈λ,y〉
over $\lambda \in R^{K}.$ We use 〈λ,y〉 to denote the standard scalar. The connection between the primal and the dual problems is explained in [11,12]. The importance of this connection is that it transforms the search over an infinite-dimensional space (the class of densities) onto the problem of finding a minimum of a convex function over a finite-dimensional space.

Below, we make use of the fact that the entropy of $\rho^{*}$ is given by:(11)$$S_{Q}\left(P^{*},y\right)=\ln Z\left(\lambda^{*}\right)+\langle \lambda^{*},y\rangle =\inf\left\{\ln Z\left(\lambda \right)+\langle \lambda ,y\rangle \right\}$$
where the label specifying the point constraint is added for emphasis and used below.

### 2.2. MEM for Data in Ranges

The extension of the procedure described in the previous section to Problem (20) goes as follows. Instead of (5), we now have
(12)$$C\left(Q,K\right)=\left\{P\in P|\;CE_{P}\left[X\right]\in K\right\}=\underset{y\in K}{⋃}C\left(Q,y\right).$$Therefore, we consider the extended entropy maximization problem:

**Problem** **5.**
*Find $\rho^{*}\left(\xi \right)$ at which $S_{Q}\left(P\right)$ achieves its maximum subject to the constraints (12). That is, determine*
$$\rho^{*}=argsup\left\{S_{Q}\left(P\right)\;|\;P\in C\left(Q,K\right)\right\}.$$


Considering (12), we can rewrite (5) as
$$\rho^{*}=argsup\left\{sup\left\{S_{Q}\left(P\right)\;|\;P\in C\left(Q,y\right)\right\},|\;y\in K\right\}=argsup\left\{S_{Q}\left(P^{*},y\right)\;|\;y\in K\right\}$$Now, making use of (11), we rewrite the last identity as
$$\sup\{\inf\left\{\ln Z\left(\lambda \right)+\langle \lambda ,y\rangle |\lambda \in R^{K}\right\}|y\in K\}.$$We can exchange the maximization with the minimization to obtain
$$\inf\{\ln Z\left(\lambda \right)+\sup\left\{\langle \lambda ,y\rangle \;|y\in K\right\}|y\in R^{K}\}.$$To compute the inner maximization, we perform the following affine transformation
$$y_{r}=\frac{K_{2}^{r}-K_{1}^{r}}{2}\zeta_{r}+\frac{K_{2}^{r}+K_{1}^{r}}{2}=\frac{K_{2}^{r}-K_{1}^{r}}{2}\left[\zeta_{r}+\frac{K_{2}^{r}+K_{1}^{r}}{K_{2}^{r}-K_{1}^{r}}\right]\;\;\;\;r=1,\ldots ,R$$
where $\zeta_{r}\in \left[-1,1\right],r=1,\ldots ,r.$ F or any $\lambda \in R^{K}$ we have
$$\sup\{\lambda_{r},\zeta_{r}|\zeta_{r}\in \left[-1,1\right]=|\lambda_{r}|.$$Therefore, we obtain that $\rho^{*}$ is as (8), that is,
(13)$$\rho^{*}\left(\xi \right)=\frac{e^{-\langle C^{t}\lambda^{*},\xi \rangle }}{Z(\lambda^{*})}$$
but this time $\lambda^{*}$ is to be found as the minimizer of
(14)$$\Sigma \left(\lambda ,K\right)=\ln Z\left(\lambda \right)+\sum_{r=1}^{R}\left(\frac{K_{2}^{r}-K_{1}^{r}}{2}\right)\left|\lambda_{r}\right|+\sum_{r=1}^{R}\left(\frac{K_{2}^{r}+K_{1}^{r}}{2}\right)\lambda_{r}.$$

### 2.3. Computation of Z(λ)

The last step in the mathematical procedure consists of the specification of the normalization factor Z(λ). In the current setup, in which the constraint space is a Cartesian product of intervals, that is, $\Omega =\prod_{i=1}^{d}\left[a_{i},b_{i}\right],$ the choice of reference measure is very simple: just set
$$dQ\left(\xi \right)=\prod_{j=1}^{d}\left(\epsilon_{a_{j}}\left(d\xi_{j}\right)+\epsilon_{b_{j}}\left(d\xi_{j}\right)\right)$$
where we use $\epsilon_{a}\left(d\xi \right)$ for the point mass at *a* (sometimes written δ(ξ−a)dξ). The intuition is that any point in $\left[a_{i},b_{i}\right]$ is the expected value with respect to a probability carried by the end points of the interval. With this, the function (9) is given by
(15)$$Z\left(\lambda \right)=\int_{\Omega }e^{-\langle C^{t}\lambda ,\xi \rangle }dQ\left(\xi \right)=\prod_{j=1}^{d}\left(e^{-{\left(C^{t}\lambda \right)}_{j}a_{j}}+e^{-{\left(C^{t}\lambda \right)}_{j}b_{j}}\right).$$

#### 2.3.1. The Case of Point Constraints

When (4) consists of determining the energy supply pattern for a given point constraint k, the dual entropy (10) now becomes
(16)$$\Sigma \left(\lambda ,k\right)=\sum_{j=1}^{d}\ln\left(e^{-{\left(C^{t}\lambda \right)}_{j}a_{j}}+e^{-{\left(C^{t}\lambda \right)}_{j}b_{j}}\right)+\langle \lambda ,k\rangle .$$This function is clearly defined, strictly convex in λ, and, since it tends to *∞* as ∥λ∥→∞, it has a unique minimizer. In addition, from the first-order condition for $\lambda^{*}$ to be a minimizer, it is easy to see that
(17)$$x_{j}^{*}=a_{j}\frac{e^{-{\left(C^{t}\lambda^{*}\right)}_{j}a_{j}}}{e^{-{\left(C^{t}\lambda^{*}\right)}_{j}a_{j}}+e^{-{\left(C^{t}\lambda^{*}\right)}_{j}b_{j}}}+b_{j}\frac{e^{-{\left(C^{t}\lambda^{*}\right)}_{j}b_{j}}}{e^{-{\left(C^{t}\lambda^{*}\right)}_{j}a_{j}}+e^{-{\left(C^{t}\lambda^{*}\right)}_{j}b_{j}}}.$$The weights of $a_{j}$ and $b_{j}$ are positive and add up to 1 and can therefore be interpreted as the (maxentropic) probability that the auxiliary random variable X assumes those values, and $x^{*}$ is its expected value. They satisfy the constraint $Cx^{*}=k$, as evidenced from the first order condition for $\lambda^{*}$ to be a minimizer.

#### 2.3.2. The Case of Data in Ranges

When the data are specified up to a range as in the previous section, the aforedescribed approach can also be used. The function Z(λ) is as above, but this time the dual entropy is different. According to the methods in Section 2 the dual entropy this time is
(18)$$\Sigma \left(\lambda ,K\right)=\sum_{j=1}^{d}\ln\left(e^{-{\left(C^{t}\lambda \right)}_{j}a_{j}}+e^{-{\left(C^{t}\lambda \right)}_{j}b_{j}}\right)+\sum_{r=1}^{R}\left(\frac{K_{2}^{r}-K_{1}^{r}}{2}\right)\left|\lambda_{r}\right|+\sum_{r=1}^{R}\left(\frac{K_{2}^{r}+K_{1}^{r}}{2}\right)\lambda_{r}.$$Again, this is a strictly convex function of λ that tends to *∞* as ∥λ∥→∞. Thus, it has a unique minimizer at some point $\lambda^{*}.$ Once this point has been found, the solution can be represented by (17), and now we have that $Cx^{*}\in K.$

## 3. The Maxentropic Solution to the Energy Diversification Problem

In this section, we see how to relabel the unknowns, describe the maxentropic solution to the resulting problem and describe how to compute the energy diversification index from the resulting solution.

### 3.1. Statement of the Problems and Representation of the Solution

#### Relabeling the Problem

Here, we relabel the unknowns and the data, not only to simplify the presentation, but also to allow the numerical implementation. We apply the following anti-lexicographic labeling ℓ:{1,2,…,N}×{1,2,…,M}→{1,2,…,d=NM} specified as follows:$$\begin{array}{c} (i,1)\to i,\;\;\;\mathrm{for}\;\;\;i=1,\ldots ,N,\;\;\&\;\;j=1\\ (i,2)\to i+N,\;\;\;\mathrm{for}\;\;\;i=1,\ldots ,N,\;\;\&\;\;j=2\\ \vdots \\ (i,M)\to i+(M-1)N,\;\;\;\mathrm{for}\;\;\;i=1,\ldots ,N,\;\;\&\;\;j=M \end{array}$$Similarly, lets us write $x_{i}\in \left[a_{i},b_{i}\right]$ to denote the constraints upon the unknowns in the new labeling. Notice that each constraint might be repeated a number of times since it is an energy production constraint and each source appears repeated as many times as its connections to demand centers.

After the relabeling of the edges of the supply-demand system, the constraint matrix C can be relabeled in the obvious way $C_{r,(i,j)}\to C_{r,\ell (i,j)}$ and our problem becomes

**Problem** **6**(Generic problem). *Find $x\in \prod_{j=1}^{d}\left[a_{j},b_{j}\right]$ such that either*
(19)Cx=k
*when we have point data or*
(20)Cx∈K.
*when the data are specified up to a range.*


In general, supply-demand problems lead to singular constraint matrices C. This is essentially due the fact that $\sum D_{i}=\sum S_{j}=T,$ which amounts to saying that there are two sets of rows in C which have the same sum, therefore they are linearly related or C is singular. As examples consider, the two matrices in Section 3.

The solution to Problems (19) and (20) is shown to be given by (17), which we display as
(21)$$x_{j}^{*}=a_{j}\frac{e^{-{\left(C^{t}\lambda^{*}\right)}_{j}a_{j}}}{e^{-{\left(C^{t}\lambda^{*}\right)}_{j}a_{j}}+e^{-{\left(C^{t}\lambda^{*}\right)}_{j}b_{j}}}+b_{j}\frac{e^{-{\left(C^{t}\lambda^{*}\right)}_{j}b_{j}}}{e^{-{\left(C^{t}\lambda^{*}\right)}_{j}a_{j}}+e^{-{\left(C^{t}\lambda^{*}\right)}_{j}b_{j}}}.$$

The difference between its applicability to (19) and (20) comes from the fact that in each case the $\lambda^{*}$ appearing in (21) is found minimizing a different (dual entropy) convex function. Observe that $x_{j}^{*}$ is a convex combination of weights adding up to 1, and therefore $x_{j}^{*}\in \left[a_{j},b_{j}\right]$ for j=1,…,d. Note that either $Cx^{*}=k$ in the case of point constraints or $Cx^{*}\in K$ in the case of extended constraints.

### 3.2. The Energy Diversification According to Stirling’s Measure

To analyze how well-diversified our numerical solutions turn out to be, we invoke Stirling’s proposal of computing the entropy of a quantity directly related to the solution, namely the energy transferred to site *i* form source *j* per unit of total demand. For this, note that the total demand $T=\sum_{i=1}^{N}D_{i}$ can also be written as $T=\sum_{\left\{i,j\right\}}x_{i,j}.$ Define now the energy transferred per unit demand by
(22)$$p_{i,i}=\frac{x_{i,j}}{T}=\frac{x_{i,j}}{\sum_{\left\{i,j\right\}}x_{i,j}}\;\;\left(i,j\right)\in \left\{1,\ldots ,N\right\}\times \left\{1,\ldots ,M\right\}.$$Conversely, using the lexicographic labeling,
(23)$$p_{k}=\frac{x_{k}}{T}=\frac{x_{k}}{\sum_{n}x_{n}}\;\;k\in \left\{1,\ldots ,NM\right\}.$$Observe that with this rescaling, $p_{k}$ are a true probability distribution.

The Stirling energy diversification coefficient is defined as the Boltzmann-Gibbs-Shannon entropy of p, namely:(24)$$ED\left(x\right)=-\sum_{k}^{NM}p_{k}\ln p_{k}.$$To check how well-diversified the maxentropic solution $x^{*}$ given by (21) is, we can compare $ED(x^{*})$ versus the most diversified probability distribution on {1,…,NM}, namely the uniform distribution which has an entropy lnNM.

## 4. Numerical Examples

Here, we implement the results obtained in the previous section in two illustrating toy examples. In the first one, we consider the simplest case of a global aggregate demand, and, in the second, we consider two possible demands: domestic and industrial.

### 4.1. Case 1: Aggregated Demand

The sources of energy, the bounds on their outputs, and their technological cost constraints are given in Table 1. All of these costs are related to the emission of combustion gases. It might be interesting to find ways of quantifying a technological (environmental) cost for the electricity produced in nuclear plants. We mention as well that one could add solar, eolic, sea wave, or geothermal energy. None of them contributes to the greenhouse effect.

We consider a uniform nonzero lower bound in all cases to accommodate the possibility that no source fails completely. Otherwise, we can set the corresponding $a_{j}$ equal to 0. The upper bounds can be thought to be given as fraction of a total possible maximal demand.

We consider two constraints, one a total demand constraint and the other an ecological or regulatory cost constraint. Thus, the constraint matrix becomes:$$C=\left[\begin{array}{cccc} 1 & 1 & 1 & 1\\ 0.0 & 0.3 & 0.7 & 0.9 \end{array}\right].$$This matrix is clearly singular. The vector of unknowns $x^{t}=\left(x_{1},x_{2},x_{3},x_{4}\right)$ is constrained to be in Ω=[0.01,0.8]×[0.01,0.5]×[0.01,0.7]×[0.01,0.4]. As point data vector, we consider $k^{t}=\left(1.8,0.5\right).$ Thus, the problem that we have to solve is
$$\mathrm{Find}\;\;\;x\in \Omega \;\;\;\mathrm{such}\;\mathrm{that}\;\;\;\left[\begin{array}{cccc} 1 & 1 & 1 & 1\\ 0.0 & 0.3 & 0.7 & 0.9 \end{array}\right]\left[\begin{array}{c} x_{1}\\ x_{2}\\ x_{3}\\ x_{4} \end{array}\right]=\left[\begin{array}{c} 1.2\\ 0.5 \end{array}\right].$$Once the optimal $\lambda^{*}$ is available, the quantities to be supplied by the sources are computed as shown in (21). The optimal solution is:$$x^{*}=\left(0.459,0.253,0.305,0.181\right)$$


**The diversification measure**


With the entropic solution to Problem (19), we form $p_{j}=x^{*}/k_{1},$ which is the proportion of the available energy supply to be provided by the *j*th supplier, and, with that, we proceed as described in Section 2.2 and compute de entropic diversification measure as:(25)$$ED\left(x^{*}\right)=-\sum_{j=1}^{4}p_{j}\ln p_{j}.$$
as proposed by Stirling, that is:$$ED\left(x^{*}\right)=-\sum_{j=1}^{4}p_{j}\ln p_{j}=1.329.$$This should be compared with the entropy of the uniform distribution, which in this case is ln4=1.386. The relative error is about 4%.

### 4.2. Case 2: Disaggregated Demand

To continue with the toy examples, we now suppose that electrical power has to be supplied to two types of consumers: domestic and industrial. Therefore, the supply of the *i*th source is split into two components $x_{i},x_{4+i},$ which means that our vector of unknowns is an eight-dimensional vector $x^{t}=\left(x_{1},x_{2},\ldots ,x_{4},\ldots ,x_{8}\right).$ The first four components describe how much of the output of the *i*th source goes into domestic demand, and the last four how much goes into industrial demand. This implies that the supply of the *i*th source has to meet the additional demand constraint $x_{i}+x_{4+i}\leq b_{i},$ where the $b_{i}$s are given in Table 1. Let us suppose that we have one technological cost constraint and one production cost constraint. The matrix of constraints is now
$$C=\left[\begin{array}{cccccccc} 1 & 1 & 1 & 1 & 0 & 0 & 0 & 0\\ 0 & 0 & 0 & 0 & 1 & 1 & 1 & 1\\ 0.0 & 0.3 & 0.7 & 0.9 & 0.0 & 0.3 & 0.7 & 0.9\\ 0.6 & 0.4 & 0.7 & 0.3 & 0.6 & 0.4 & 0.7 & 0.3\\ 1 & 0 & 0 & 0 & 1 & 0 & 0 & 0\\ 0 & 1 & 0 & 0 & 0 & 1 & 0 & 0\\ 0 & 0 & 1 & 0 & 0 & 0 & 1 & 0\\ 0 & 0 & 0 & 1 & 0 & 0 & 0 & 1 \end{array}\right].$$Note that the third and fourth rows consist of two equal halves. The third contains the technological (environmental) cost constraint and the fourth contains the production cost constraint. Note as well that the sum of Rows 1 and 2 equals the sum of Rows 5–8, therefore the determinant of C is 0. That is, even though the matrix C is square, the problem is nevertheless ill-posed.

Let us now suppose that the aggregate domestic and industrial demands are, respectively, $k_{1}=0.9$; $k_{2}=1.05.$ This time the total is intentionally higher than before. Let the technological constraint upon the energy production be $k_{3}=0.73$ while the production constraint is $k_{4}=1.06.$ Now, instead of point data for the remaining constraints, we have $x_{i}+x_{4+i}.$ Again, we are supposing that every supplier produces more than a minimum $a_{i}$ and less than $b_{i}.$ The range for the data in this example is the following:$$K=\left\{k_{1}\right\}\times \left\{k_{2}\right\}\times \left\{k_{3}\right\}\times \left\{k_{4}\right\}\times \left[K_{1}^{5},K_{2}^{5}\right]\times \left[K_{1}^{6},K_{2}^{6}\right]\times \left[K_{1}^{7},K_{2}^{7}\right]\times \left[K_{1}^{8},K_{2}^{8}\right].$$For the numerical computations, we use:K={0.9}×{1.05}×{0.73}×{1.06}×[0.65,0.75]×[0.35,0.45]×[0.65,0.75]×[0.25,0.35].

Clearly, some of the ranges are degenerate (just a point) while the others are just the natural range for the corresponding supply. We still have to specify the range for the components of x. The obvious choice is $x_{i},x_{4+i}\in \left[a_{i},b_{i}\right],$ which suggests the following choice for the constraint (or sample) space Ω:$$\begin{array}{c} \Omega =[0.01,0.8]\times [0.01,0.7]\times [0.01,0.5]\times [0.01,0.4]\\ \times [0.01,0.8]\times [0.01,0.7]\times [0.01,0.5]\times [0.01,0.4]. \end{array}$$Note that we are assuming that the constraints upon the individual demands are the same as the constraint upon the output of each source. This facilitates the typography.

The problem to solve now becomes:Findx∈Ωsuch thatCx∈K.

As the previous case, the solution is also given by
$$x_{j}^{*}=a_{j}\frac{e^{-{\left(C^{t}\lambda^{*}\right)}_{j}a_{j}}}{e^{-{\left(C^{t}\lambda^{*}\right)}_{j}a_{j}}+e^{-{\left(C^{t}\lambda^{*}\right)}_{j}b_{j}}}+b_{j}\frac{e^{-{\left(C^{t}\lambda^{*}\right)}_{j}b_{j}}}{e^{-{\left(C^{t}\lambda^{*}\right)}_{j}a_{j}}+e^{-{\left(C^{t}\lambda^{*}\right)}_{j}b_{j}}}\;\;\;j=1,\ldots ,8.$$In this case, C is the 8×8-matrix specified above. The vector $\lambda^{*}$ is an eight-dimensional vector that is obtained minimizing the dual entropy (18)
$$\Sigma \left(\lambda ,K\right)=\sum_{j=1}^{8}\ln\left(e^{-{\left(C^{t}\lambda \right)}_{j}a_{j}}+e^{-{\left(C^{t}\lambda \right)}_{j}b_{j}}\right)+\sum_{r=1}^{8}\left(\frac{K_{2}^{r}-K_{1}^{r}}{2}\right)\left|\lambda_{r}\right|+\sum_{r=1}^{8}\left(\frac{K_{2}^{r}+K_{1}^{r}}{2}\right)\lambda_{r}.$$

When we separate the degenerate from the non-degenerate intervals, the second and third summation become:$$\sum_{i=1}^{4}\lambda_{i}k_{i}+\sum_{r=5}^{8}\left(\frac{K_{2}^{r}-K_{1}^{r}}{2}\right)\left|\lambda_{r}\right|+\sum_{r=5}^{8}\left(\frac{K_{2}^{r}+K_{1}^{r}}{2}\right)\lambda_{r}.$$Undoing the lexicographic ordering, the maxentropic solution in this case is
$x_{i,j}$123410.3000.1780.3010.10620.3690.2210.3270.119


**The diversification measure**


Now that we have obtained the $x_{j}^{*}$ solving our problem, we apply (23), form $p_{j}=x^{*}/\left(k_{1}+k_{2}\right),$ and compute the entropic measure of diversity (24).
$$ED\left(x^{*}\right)=-\sum_{j=1}^{8}p_{j}\ln p_{j}=-\sum_{i,j}\frac{x_{i,j}^{*}}{k_{1}+k_{2}}\ln\frac{x_{i,j}^{*}}{k_{1}+k_{2}}=2.00.$$Taking into account that the maximum diversification possible corresponds to a uniform probability distributions $q_{j}=1/8$ and that ln8=2.079, the maxentropic solution is well diversified.

## 5. Conclusions and Policy Implications

The two illustrating toy examples considered show the viability of the method of maximum entropy in the mean to solve constrained energy transportation problems. To test the diversification of the transportation policies obtained, we rescaled the solution $x^{*}$ to the energy transportation problem. That rescaling can be interpreted as the energy transported per unit of total demand. It renders the solution as a probability distribution on the edges of the transportation graph. The resulting entropy of the probability is a measure of diversification.

An interesting aspect of the method of maximum entropy in the mean is that policy requirements can be imposed as ab initio constraints on the solution.

In the two examples that we considered, the solution happened to be quite well diversified, but a general comparison is still lacking.

## Figures and Tables

**Table 1 entropy-23-00281-t001:** Toy dataset.

Type	Source	Lower Bd. (*a*)	Upper Bd. (*b*)	Cost
1	Hydraulic	0.01	0.8	0.00
2	Gas	0.01	0.5	0.37
3	Fuel	0.01	0.7	0.76
4	Coal	0.01	0.4	0.96

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
