# Peer review of "Electrical Power Diversification: An Approach Based on the Method of Maximum Entropy in the Mean"

_entropy, 2021, doi:10.3390/e23030281_

Round 1

Reviewer 1 Report

Dear Editor,

With the energy-distribution process emerges a problem with several edges; these are, ecological, technological or economic cost constraints. The authors present an interesting approach using the method of maximum entropy in the mean. 

I agree with the presentation of the author. However, I have the only point. Namely, authors devoted the section "Appendix" to discuss the definition pertinent to the paper, prominently. I encourage the author to put the Appendix as a Section of the paper.

After taking into account the present suggestion I recommend to publish this work.

Author Response

Dear referee,

We thank you for your insightful suggestions. We relocated the Appendix as the new Section 3.

In Section 1.2 we deleted part of the first paragraph and added a new paragraph to explain the contents of Section 3 (formerly the appendix) It is highlighted in blue.

Also, words changed here or there appear now highlighted in blue.

Best regards,

Enrique ter Horst (on behalf of all coauthors)

Reviewer 2 Report

Review of 'Electrical power diversifi cation: An approach
based on the method of maximum entropy in the mean'

The authors formulate models of electricity supply-demand, with increased complexity. The basic model is fi rst expanded to take into account
the environmental impact, and then the constraints are relaxed to allow interval values. The resulting inverse problems are solved by the Maximum
Entropy in the Mean (MEM) method. MEM is described in the Appendix.
Workings of MEM are illustrated by two interesting examples.

The paper is clearly written, concise, self-contained; a pleasure to read.

Author Response

Dear referee,

We thank you for your insightful suggestions. We have done a thorough spell checking and words changed here or there appear now highlighted in blue.

Best regards,

Enrique ter Horst (on behalf of all coauthors)
